# A controlled study of emotional dysfunction in adult women with ADHD

**Ortal Slobodin[1]\*, Michal Har Sinay[2], Ada H. Zohar[2]**

**1** School of Education, Ben-Gurion University of the Negev, Beer-Sheva, Israel, **2** Department of Behavioral Sciences, Clinical Psychology Graduate Program, Ruppin Academic Center, Emek Hefer, Israel

\* Ortal.slobodin@gmail.com

## Abstract

### Objective

Given the increased socio-emotional burden of ADHD symptoms in women, the current study examined the associations between ADHD symptoms, emotional dys-regulation, and executive function deficits. We also examined two types of executive function deficits, working memory and task shifting, as potential mediators of the link between women's ADHD symptoms and emotional dysregulation.

### Method

The study included 176 women between 20 and 30 years of age. Of these, 82 reported a known diagnosis of ADHD, and 94 reported no known diagnosis. Participants completed questionnaires addressing the severity of ADHD symptoms, executive function deficits, emotional dysregulation, alexithymia, positive affect, and negative affect.

### Results

Emotional dysregulation was associated with ADHD symptom severity and executive function deficits. Women with ADHD showed more frequent use of non-adaptive emotion regulation strategies, including negative affect and alexithymia, compared to women without ADHD symptoms. In addition, results showed that executive function deficits (in working memory and task shifting) mediated the relationship between ADHD symptoms and emotional dysregulation.

### Conclusions

These findings add further support to the growing body of research suggesting that not only inattention, hyperactivity, and impulsivity, but also emotional dysregulation are core components of ADHD. Shifting clinical attention to emotional dysregulation symptoms may improve the diagnosis and treatment of women's ADHD.

**Data availability statement:** Data associated with this manuscript are available in FIGSHARE data repository. Doi: https://doi.org/10.6084/m9.figshare.30608699.

**Funding:** The author(s) received no specific funding for this work.

## Introduction

Emotional dysregulation is highly prevalent in children, adolescents, and adults with ADHD [1,2] and is associated with a host of adverse outcomes, especially in women [3]. Less is known, however, about the neurocognitive and behavioral mechanisms that underlie the association between ADHD and emotional dysregulation [4]. Research suggests that because emotional regulation involves strategies like inhibition, reappraisal, and suppression, emotional dysregulation in ADHD might be closely related to executive function deficits [5]. Based on previous studies that associated both working memory and task shifting deficits with emotional dysregulation in children with ADHD [6,7], we examined the role of these two executive function deficits (i.e., working memory, task shifting) in the relationship between women's ADHD symptoms and emotional regulation deficits.

### The gender gap in ADHD

ADHD is one of the most prevalent childhood disorders, with a global prevalence of 2–11% [8]. In most cases, ADHD symptoms persist in adulthood, affecting more than 3–4% of the adult population [9]. Adult ADHD core symptoms (i.e., inattention, hyperactivity, and impulsivity) are associated with functional, emotional, academic, cognitive, family, health, and occupational impairments [10].

For many years, ADHD was considered a predominantly male condition [11]. A study based on 13 meta-analytic systematic reviews (588 primary studies, 3,277,590 participants) showed that the ADHD prevalence was twice higher in boys (10%) compared to girls (5%) [9]. Similarly, recent global data suggested that the incidence and prevalence of ADHD were 2.57 and 2.46 times higher in males than in females [12]. The gender gap in ADHD prevalence reflects a different manifestation of ADHD (e.g., males present more hyperactive/impulsive symptoms, whereas females present more inattentive symptoms, [13]), as well as systematic referral and identification biases [14,15]. Previous studies have consistently shown that teachers, parents, and professionals are more likely to recognize ADHD-related symptoms in boys than in girls and are more likely to refer boys to treatment [16,17]. These findings suggest that gender bias in recognizing ADHD in girls and women may reflect limited familiarity with the female ADHD phenotype among parents, teachers, and professionals, but also raise questions about whether current diagnostic criteria for ADHD are sufficiently sensitive to detect ADHD in females [18].

### ADHD presentation in women

Because most research on ADHD has traditionally been focused on male samples, there is a lack of knowledge of ADHD in females, both in research and clinical practice [19]. Existing studies, however, reported notable gender differences in ADHD symptom phenomenology [20]. A recent review [21] pointed to a higher prevalence of inattention, emotional dysregulation, and comorbid conditions (e.g., mood and anxiety disorders) in women with ADHD compared with men. The predominance of the inattentive presentation in females and the lower rates of co-occurring disruptive

behavior disorders suppress the visibility of female symptoms and complicate identification and diagnosis [22]. Females are also more likely to present the inattentive ADHD subtype, which is more difficult to identify by parents or teachers compared to the combined or hyperactive-impulsive subtypes [14,23].

Additional studies suggested that women with ADHD present higher rates of depression, borderline personality disorder [24], somatic symptoms, eating disorders [25], and suicidality [26] than men. A key finding in both qualitative [27] and quantitative [28] studies, is that women with ADHD report significantly greater social and interpersonal difficulties than males with ADHD, and that this social dysfunction has more adverse consequences for women [29]. ADHD-related socio-emotional difficulties may be especially impactful among women [28] due to gender role socialization with the expectation that girls and women take primary responsibility for others, and avoid aggressive or oppositional behavior [3]. ADHD-related socio-emotional difficulties seem to be consistent with the gender paradox, which posits that individuals with gender atypical disorders, such as females with ADHD, manifest greater and more varied difficulties compared to individuals with gender-typical disorders (i.e., men with ADHD; [30]).

The transition to adulthood poses new challenges for women with ADHD [31]. Whereas ADHD symptoms in youth are dominant in home and school settings, adults have many more responsibilities, which create more domains for ADHD-related dysfunction [3]. Establishing positive occupational, peer, and family relationships involves both executive aspects (e.g., planning, flexibility, organization) and emotional regulation [32].

## Emotional dysregulation and ADHD

Emotional regulation can be defined as a complex process by which people modulate their emotions to direct their behavior towards goals using different strategies, such as cognitive reappraisal and response inhibition [33]. Symptoms of emotional dysregulation, such as increased negative emotionality, heightened emotional reactivity, alexithymia, and maladaptive emotion regulation strategies [34,28], are common and persistent in youth and adults with ADHD and are associated with symptom severity, increased social conflict, psychiatric comorbidities, and criminality [3,2]. While most studies on emotional dysregulation and ADHD have overlooked the role of gender, a few studies showed that women with ADHD presented greater emotional dysregulation [35,36]. Theoretically, three models have been conceptualized to explain the overlap between emotion dysregulation and ADHD [37]. According to the first model, emotional dysregulation is a core symptom of ADHD due to joint neurocognitive deficits in self-regulation that underline both the cognitive and emotional regulation systems [38]. This joint deficit could be a common result of the regulation of physiological arousal, inhibition deficits, attention regulation, and action planning [39]. The second model perceives ADHD with emotional dysregulation as qualitatively different from pure ADHD, which has a distinct etiological entity and ADHD course [40,41]. The third model, supported by findings of moderate correlations between ADHD symptoms and emotional dysregulation, posits that ADHD and emotional dysregulation may share overlapping circuits, but still have distinct neurocognitive deficits [42]. Understanding and mitigating emotion dysregulation in ADHD is imperative because it increases the burden of illness associated with ADHD than ADHD symptoms alone [43]. However, research on the subject is still very limited (for a systematic review, please see [2]). There are several reasons to study emotional dysregulation in women with ADHD. First, emotional difficulties are common symptoms with adverse consequences for women with ADHD. Second, emotional problems and internalizing symptoms [44], which often become prominent during the transition from adolescence into adulthood, may contribute to women's delayed ADHD diagnosis. Finally, emotional dysregulation may be a transdiagnostic mechanism that explains the overlap between ADHD and several common comorbidities, such as depression [24], eating disorders [45], and borderline personality disorder [46].

## ADHD and executive function

Attention is an executive function as is the regulation of impulsivity and hyperactivity. Thus, it is likely that individuals diagnosed with ADHD will show deficits in executive functions. Barkely [47] goes a step further by defining ADHD as an executive functions disorder. In this study, we chose to measure two elements of executive function that are not closely

related to the diagnostic symptoms of ADHD or to emotional regulation: working memory and task shifting. Individuals with ADHD often exhibit significant deficits in executive functions which are not part of the core disorder, in particular deficits in working memory and task shifting [48]. Working memory involves the temporary storage and manipulation of information, while task shifting refers to the ability to switch between tasks or mental sets with flexibility [49]. These are key cognitive functions for goal-directed behavior [50]. Deficits in working memory are associated with difficulties in holding and processing information over short periods, while impairments in task shifting contribute to problems with cognitive flexibility and the ability to adapt to new tasks or situations [51]. We measured these two functions in the current study to examine the possibility that they contribute to difficulties in emotion regulation over and beyond the contribution of core ADHD symptomatology.

## Emotional regulation and executive functions in ADHD

Emotional regulation involves the inhibition of negative affect expressions [52], cognitive reappraisal, and suppression strategies [53]. Therefore, it might be closely linked to executive functions, such as inhibition, attention, effortful control, working memory, and shifting [5]. Studies in children with ADHD have associated emotion regulation difficulties with deficits in working memory [7], inhibitory control [54], and task shifting [6]. Most studies examining these relations in children have investigated executive functions in isolation rather than determining whether specific executive functions uniquely predict emotion dysregulation [55]. An exception is the study of Sjowall et al. [6], which showed that children with ADHD exhibited worse executive functioning and emotion regulation than children without ADHD and that all three executive functions (i.e., working memory, inhibition, shifting) were significantly correlated with emotion dysregulation. Additionally, Groves et al. [55] indicated that better-developed working memory, but not inhibitory control or set shifting, predicted fewer ADHD symptoms and better emotion regulation skills. In adults, research on the relationship between emotion regulation and executive functions in ADHD is limited and has revealed inconsistent findings [56]. For example, Rosello et al. [57] found that severe impairment in self-regulation was associated with executive function deficits, higher levels of ADHD symptoms, increased comorbidity, and worse quality of life. In contrast, Liu et al. [58] showed that ADHD was associated with more expressive suppression, probably due to compensation mechanisms, and that reduced use of cognitive reappraisal accounts for emotion dysregulation in ADHD.

The close relationship between executive functions and emotional regulation in ADHD as well as in other disorders, such as antisocial and borderline personality disorders, has led some scholars to suggest that emotional dysregulation may be understood as an integral part of impaired executive functioning. Van Cauwenberge et al. [59], for instance, found that children with ADHD exhibit a generic interference control deficit, rather than specific emotional deficits. Others, however, found that emotion regulation deficits in ADHD are associated with both specific and general self-regulation deficits traceable on the level of neuropsychological, neurophysiological, and psychophysiological assessments [28].

## The current study

The current study examined the associations between women's ADHD symptoms, emotional dysregulation, and executive function deficits. We also examined two types of executive function deficits, working memory and task shifting, as potential mediators of the link between women's ADHD symptoms and emotional dysregulation.. In particular, we examined the following hypotheses: H1. Symptoms of ADHD will be correlated with difficulties in emotional function, as well as with executive function deficits. H2. Women with ADHD will show greater impairment in various aspects of emotional function compared to women without ADHD. H3. Difficulties in executive function, specifically working memory and task shifting, will add explained variance to emotion dysregulation above and beyond the contribution of ADHD symptoms. H4. Difficulties in executive functions will mediate the relationship between symptoms of ADHD and emotional dysregulation.

## Methods

We report how we determined our sample size, all data exclusions, all manipulations, and all measures in the study.

### Procedure

To test the feasibility of ascertaining women with a known diagnosis of ADHD, we posted a request for women between 20 and 30 years of age who were in one of two dedicated Facebook groups: a group for students who want to be accepted into a competitive graduate degree, and a group for women who live in a large urban area. The post explained that the study examines differences between women with and without a known diagnosis of ADHD and a matched control group and asked interested women with a known ADHD diagnosis to contact the research team via email. This group was offered a small gift voucher in return for participation. Women who contacted the research team received an online questionnaire and were asked to complete it [60]. The questionnaire was completed by 51 participants with a known ADHD diagnosis. We then offered undergraduate women enrolled in an introductory psychology class to participate in the study in return for academic credit. This group yielded 31 young women with a known ADHD diagnosis and 94 with no known diagnosis. The research protocol was approved by the IRB of the responsible academic institute (#223/2023). All participants signed a written informed consent, which was approved by the IRB. To assess participants' capacity to sign informed consent, we used Biros' [61] proposed method for evaluating attitudes toward research involvement. Because we used a community sample of undergraduate students and women who were active in social media and volunteered to participate in the survey, we assumed their full capacity to provide a meaningful decision regarding their research participation, and that informed consent could be obtained directly from potential participants. Data were collected between 29/1/24 and 12/5/24.

### Participants

The study included 176 women between the ages of 20 and 30. All participants were undergraduate students or recent graduates. Of these, 82 reported a known diagnosis of ADHD, and 94 had reported no known diagnosis. All participants also completed a self-report on ADHD symptoms, used for diagnosing ADHD in adults in the community. The screening questionnaire was validated in the past [62] against a clinical interview and a cut-off point for optimal sensitivity and specificity was then defined.

In the current study, we were interested in understanding the relationship between ADHD, executive function deficits, and emotional dysregulation. The operationalization of ADHD included two different measures. The first measure was a self-reported lifetime ADHD diagnosis status (i.e., whether the participant has ever been diagnosed with ADHD by a health care professional), and the second was the Adult ADHD Self-Report Scale (ASRS; [63]), a well-established scale of ADHD symptoms. Although the ASRS is considered a valid measure of ADHD, higher scores are not always diagnostic [64]. For example, Brevik et al. [65] found that a high score on the ASRS had, at best, only a 22% chance of accurately identifying those with true ADHD, meaning that a large proportion of those who truly had ADHD were not correctly identified. At the same time, relying exclusively on self-reports of ADHD diagnosis may exclude many women who were never officially diagnosed despite having extensive ADHD symptoms [66]. Based on these two measures of ADHD, we identified three distinct groups of participants. The first group of women ($N=70$) reported a known diagnosis of ADHD and scored above the ASRS cut-off point. The second group included women without a known diagnosis of ADHD and scored below the ASRS cut-off point ($N=64$). The final group of participants included women for whom the diagnostic status and current symptomatic level according to the ASRS were not in agreement ($N=42$). Of them, 12 women reported a lifetime ADHD diagnosis but scored below the clinical ASRS cut-off point, and 30 women did not report a known diagnosis of ADHD but scored above the cut-off point. We have termed this third group "probable ADHD". As seen in Table A in S1 Text, no significant differences were observed between the three groups in age, years of education, and employment rates. However,

as described below, the three study groups significantly differed on most study variables, including ADHD symptom level, working memory deficits, task shifting deficits, and emotional dysregulation. These differences support the view that the three research groups represent different levels or forms of impairment. These findings are further elaborated in the results section.

### Measures

**Background variables.** These variables included age, years of education, and employment status.

**ADHD symptoms.** The Adult ADHD Self-Report Scale (ASRS; [63]) was used to assess ADHD symptoms, using the Hebrew version [67]. The questionnaire includes 18 items divided into two subscales: inattentiveness, e.g., "How often do you have difficulty keeping your attention when you are doing boring or repetitive work?" and hyperactivity/impulsivity, e.g., "How often do you find yourself talking too much when you are in social situations?". The items are measured on a Likert scale ranging from 1 (never) to 5 (very often). Each participant's score is calculated as the sum of items, where a score above 51 indicates a high probability of ADHD. In the current study, the reliability of the entire questionnaire was.90, and the reliability of the subscales were.86 and.86, respectively.

**Self-reported lifetime ADHD diagnosis status.** Participants answered the following questions: (1) Have you been diagnosed with ADHD; (2) At what age; (3) By which professional? (Choose from options: psychologist, neurologist, psychiatrist, psycho-didactic assessment, other);.(4) Did you get treatment for ADHD in the past or present; (5) What kind of treatment? (Choose from options: medication treatment, other); (6) Are you currently under the influence of ADHD medication?

**Executive functions.** Executive functions were measured using the Behavior Rating Inventory of Executive Function-Adult (BRIEF-A; [68]) in the Hebrew version [69]. In this study, we used the subscales found to be particularly important for ADHD. Therefore, two subscales were included: shift (six items), e.g., "I have trouble thinking of a different way to solve a problem when stuck" and working memory (eight items), e.g., "I have trouble with jobs or tasks that have more than one step ". Items are measured on a Likert scale ranging from 1 (never) to 3 (often). Each participant's score is calculated as the sum of items, where a higher score indicates lower executive functions. The reliability of the subscales was.74 and.86, respectively.

**Emotion regulation.** Emotion regulation was measured using the shortened version of the Difficulties in Emotion Regulation Scale 16 (DERS-16; [70]) in the Hebrew version [71]. The questionnaire includes 16 items divided into five subscales: non-acceptance of emotional responses, difficulties engaging in goal-directed behavior, impulse control difficulties, limited access to emotion regulation strategies, and lack of emotional clarity. Items are measured on a Likert scale ranging from 1 (almost never) to 5 (almost always). Each participant's score is calculated as the sum of items, where a higher score indicates greater difficulty in emotion regulation. In the current study, the reliability of the entire questionnaire was.92, and the reliability of the subscales ranged between.79−.89, after deciding to remove the "lack of emotional clarity" scale due to its low reliability of.57.

**Alexithymia.** Alexithymia was measured using the Toronto Alexithymia Scale (TAS-20; [72]). The questionnaire aims to measure the inability to identify, name, and attribute emotions. The questionnaire consists of 20 items divided into three subscales: difficulty identifying feelings (6 items), difficulty describing feelings (5 items), and externally-oriented thinking (9 items). Items are measured on a Likert scale ranging from 1 (strongly disagree) to 5 (strongly agree). Each subscale is calculated as the sum of its items, resulting in three scores. Higher scores indicate greater difficulty in identifying, naming, and attributing emotions. In the current study, the reliability of the subscales was.88,.80, and.59, respectively, and the reliability of the entire questionnaire was.85. Due to the low Cronbach's alpha of the externally-oriented thinking subscale, (also documented in previous use of the Hebrew version of the scale; [62]), we removed the nine relevant items from the analysis. The total reliability of the Alexithymia scale after removing the externally-oriented thinking subscale was.90.

**Positive and negative affect.** Affect was measured using The Positive and Negative Affect Scale (PANAS; [73]). The questionnaire includes 20 items divided into two subscales: positive emotions, e.g., "excited" and negative emotions, e.g., "upset". Items are measured on a Likert scale ranging from 1 (very slightly or not at all) to 5 (extremely). Each subscale is calculated as the sum of its items, resulting in two subscale scores. Higher scores indicate higher levels of emotion. In the current study, the reliability of the positive affect scale was.83 and the reliability of the negative affect scale was.84.

## Data analysis

The data was downloaded from Qualtrics into SPSS, and all analyses were conducted in SPSS version 27. To examine the first hypothesis, Pearson correlations were computed. To examine the second hypothesis, we conducted a Multiple Analysis of Variance (MANOVA) with post-hoc comparisons using the Bonferroni correction. To examine the third hypothesis, we conducted a hierarchical linear regression analysis. To explore the fourth hypothesis, we conducted conditional process modeling, using the PROCESS macro model 4 with the bootstrapping procedure [74]. Power calculations revealed that for multiple regression with four predictors and a power of 0.8, a minimum of 87 participants was required [75].

## Results

Table B in S1 Text presents normality and homogeneity for the study variables. Levene's Test for equality of variances was not significant, indicating that the assumption of homogeneity of variances was met. Skewness values were all between −0.5 and 0.5, and Kurtosis values were between −0.72 and 1.03. Therefore, symmetry and normality could be determined [76]. Also, Variance Inflation Factor (VIF) values indicated no issues with multicollinearity, with all predictors showing VIFs below 2.5. Specifically, VIFs ranged from 1.03 to 2.47.

*Hypothesis 1*: Symptoms of ADHD will be correlated with difficulties in emotional function, as well as with executive function deficits. As can be seen in Table C in S1 Text, ADHD symptoms were positively and significantly correlated with all aspects of emotional dysfunction and difficulties with executive function; but not with positive affect.

*Hypothesis 2:* Women with ADHD will show greater impairment in various aspects of emotional function compared to women without ADHD.

Although the research design called for comparing two groups those with and without a known diagnosis of ADHD, our data included an intermediate group of women with a probable diagnosis of ADHD, either having a known diagnosis but currently reporting ADHD symptoms (as measured by the ASRS) below the clinical cut-off point, or having no known diagnosis but scoring above the clinical cut-off point for ADHD on the ASRS. Thus, we compared three groups: Group 1: women with ADHD ($N=70$); Group 2: with a probable diagnosis ($N=42$); and Group 3: those without ADHD ($N=64$). Multiple Analysis of Variance comparing the three groups was significant for all study variables: ADHD symptoms as measured by the ASRS $F_{(1,175)} = 164.26$, $p<.001$; Working memory deficit as measured by the BRIEF-A: $F_{(1,175)} = 79.28$, $p<.001$; Deficits in Task Shifting as measured by the BRIEF-A: $F_{(2,176)} = 33.74$, $p<.001$; Emotional dysregulation as measured by the DERS: $F_{(1,175)} = 22.17$, $p<.001$; Positive affect as measured by the PANAS: $F_{(1,175)} = 3.75$, $p<.001$; and Negative affect as measured by the PANAS: $F_{(1,175)} = 9.55$, $p<.001$. Post-hoc comparisons with Bonferroni corrections showed that the three study groups significantly differed from each other on most study variables. Specifically, women with ADHD (group 1) showed significantly higher levels of ADHD, working memory deficits, task shifting deficits, and emotional dysregulation, compared with women with probable ADHD (group 2) and those without ADHD (group 3). Women with ADHD also showed higher levels of alexithymia and negative affect than the other two groups and lower positive affect than women with probable ADHD. Additionally, women with probable ADHD showed significantly higher levels of ADHD, working memory deficits, task shifting deficits, and emotional dysregulation compared with women without ADHD. Results are presented in Fig A in S1 Text.

*Hypothesis 3*: Difficulties in executive function, specifically working memory and task shifting, will add explained variance to emotion dysregulation above and beyond the contribution of ADHD symptoms. To address whether executive

function deficits will predict emotional dysregulation above and beyond ADHD symptoms, we computed two linear regression analyses (Table D in S1 Text). As can be seen, working memory deficits accounted for a significant amount of variance in emotional dysregulation scores $F_{(4,151)}$ = 21.63, $p < .001$, $R^2 = .37$, $R^2$ change = .05, over and above the contribution of age, education, and ADHD symptoms, $F_{(3,151)}$ = 23.40, $p < .001$, $R^2 = .32$, $R^2$ change = .31. The same regression was repeated using task-shifting deficits. In this case, task shifting deficits accounted for a significant and reasonably large amount of variance in emotional dysregulation scores $F_{(4,151)}$ = 27.00, $p < .001$, $R^2 = .42$, $R^2$ change = .10, over and above the contribution of age, education, and ADHD symptoms, $F_{(3,151)}$ = 23.40, $p < .001$, $R^2 = 0.32$, $R^2$ change = .31.

*Hypothesis 4:* Difficulties in executive functions will mediate the relationship between symptoms of ADHD and emotional dysregulation. To test whether working memory and task shifting, independently, mediate the relation between ADHD symptoms and emotional dysregulation, we used Hayes's [74] bootstrap method. We computed two multiple mediation models testing whether working memory and task-shifting deficits had an indirect effect on the relationship between ADHD and emotional dysregulation. In each equation, we covaried age and education. Results are presented in Fig B in S1 Text.

First, we examined the mediating role of working memory deficits on the relationship between ADHD symptoms and emotional dysregulation. The total model was significant, [$F_{(4,147)}$ = 21.62, $p < .001$]. accounting for 37% of the variance in emotional dysregulation. ADHD symptoms had a positive main effect on emotional dysregulation ($\beta = .30$, $p = .004$) and so did working memory deficits ($\beta = .34$, $p = .001$). Consistent with our hypothesis, the indirect effect of working memory deficits on emotional dysregulation was significant ($\beta = .27$, lower limit confidence interval [LLCI]= 0.12, upper limit confidence interval [ULCI] = 0.42), indicating its mediating role in the relationship between ADHD symptoms and emotional dysregulation. Given that the direct effect of ADHD on emotional dysregulation was also significant ($\beta = .02$, $p = .004$), a partial mediation is supported.

Next, we examined the mediating role of task shifting deficits on the relationship between ADHD symptoms and emotional dysregulation. The total model was significant [$F_{(4,147)}$ = 27.00, $p < .001$], accounting for 42% of the variance in emotional dysregulation.

ADHD symptoms had a positive main effect on emotional dysregulation ($\beta = .36$, $p < .001$) and so did task shifting deficits ($\beta = .38$, $p < .001$). Consistent with our hypothesis, the indirect effect of task shifting deficits on emotional dysregulation was significant, $\beta = .21$, lower limit confidence interval [LLCI]= 0.12, upper limit confidence interval [ULCI] = 0.30, indicating its mediating role in the relationship between ADHD symptoms and emotional dysregulation. Given that the direct effect of ADHD on emotional dysregulation was also significant ($\beta = .03$, $p < .001$), a partial mediation is supported.

## Discussion

Given accumulating evidence about the increased socio-emotional burden of ADHD symptoms in women [3], the current study examined the associations between women's ADHD symptoms, emotional dysregulation, and executive function deficits. We also examined two types of executive function deficits, working memory and task shifting, as potential mediators of the link between women's ADHD symptoms and emotional dysregulation. Consistent with previous findings in children [77], adolescents [78], and adults [34], our findings showed that emotional dysregulation was associated with ADHD symptom severity and executive functioning impairment. We also found that the frequency of using non-adaptive emotion regulation strategies, including negative affect and alexithymia, distinguished between women with diagnosed ADHD, undiagnosed women with clinical levels of ADHD, and controls. These findings support the growing body of research suggesting that not only inattention, hyperactivity, and impulsiveness, but also emotional dysregulation are core components of ADHD [2].

In addition to the direct effect of ADHD symptoms on emotional dysregulation, ADHD symptoms had an indirect effect on emotional dysregulation via executive function impairments (working memory and task-shifting deficits). These results, together with the finding that executive functioning impairments contribute to emotional dysregulation above and beyond ADHD symptoms alone, suggest that emotion dysregulation in ADHD may reflect both an affective outcome of ADHD symptomatology and an outcome of impaired working memory and task shifting [4].

Although studies on the relationship between emotion regulation and executive functions in ADHD are limited, our findings are consistent with previous literature that demonstrates that working memory predicts ADHD symptom severity both cross-sectionally [79[ and longitudinally [80], as well as research demonstrating that working memory capacity predicts emotion regulation skills in children with ADHD [4,55]. One possible explanation for these associations is that emotional dysregulation may be understood as an integral part of impaired executive functioning [50]. Van Cauwenberge et al. [59], for instance, found that children with ADHD exhibit a generic interference control deficit, rather than specific emotional deficits. Other theories of ADHD emphasize the inherent contribution of emotional dysregulation to executive failure, suggesting constructs such as emotional impulsivity as another core symptom of the disorder [81].

Our results also indicated that although women with ADHD reported higher levels of emotional dysregulation, alexithymia, and negative affect than women without ADHD, the two groups did not differ in their levels of positive affect. Studies addressing positive affect in ADHD has revealed mixed results. Breaux et al. [82] showed that children with ADHD showed, on average, lower levels of positive affect, and higher levels of negative affect compared to children without ADHD. Importantly, children with ADHD also showed greater affect variability (changes in affect within the day or across days), which was associated with worse internalizing and externalizing symptoms and social functioning impairment. However, other studies suggested that ADHD, when not coupled with other comorbidities, might be related to increased positive affect [83]. One possible explanation for these unexpected findings is related to the type of items included in the PANAS. Because some positive affect items at least partly reflect higher arousal or activity levels, particularly in the context of ADHD (excited, energetic, active, lively), ADHD symptoms and positive affect may be highly associated [83]. Another possible explanation is that children and adults with ADHD are likely to be engaged in a positive illusory bias, namely, a disparity between self-report of competence and actual competence such that self-reported competence is substantially higher than actual competence [84]. Engaging in positive illusory bias may lead to an under-report of ADHD related difficulties and over-report on well-being due to having overly positive views of oneself [85]. It is also possible that people with ADHD differ from those without ADHD in the ways they experience and express negative and positive emotions. For example, Eadeh et al. [86] found that negative and positive affect are more strongly negatively associated in adolescents without ADHD, whereas among individuals diagnosed with ADHD, the experience of positive affect seems to be less likely to exclude an experience of negative affect. These findings call for future research that will focus on positive and negative affect in women with ADHD and their relationship with symptom level and social functioning.

Our results should be considered in the light of several limitations. First, the self-selected sampling method may limit our ability to generalize our findings to the broader population of women with ADHD. The current study included a relatively limited sample of non-referred, educated women who volunteered to participate in the study. Therefore, they may not represent women with more extreme levels of ADHD symptoms and functional impairment [87].

Second, we have measured executive function deficits with self-report measures rather than performance-based tests. While we used well-validated and widely used scales, self-report measures of executive function are sensitive to reported bias, such as underreporting [88], and may correlate with other variables, such as personality traits [89]. Moreover, reliance on self-report measures may involve problems of shared method variance, such that the associations obtained between women's ADHD symptoms, emotional dysregulation, and executive function impairment may become artificially inflated [Liu et al. 2016]. To address this concern, we used both self-report of ADHD lifetime diagnosis and the ASRS. We also tested for multicollinearity and found no issues. More objective measures, such as physiological indicators of emotional reactivity and performance-based tests evaluating executive functions, might have yielded different outcomes and should be considered in future studies [55]. Third, we did not assess potential comorbid conditions, which may have contributed to emotional dysfunction. Anxiety and depression, in particular, are highly prevalent comorbid conditions in ADHD [90], especially in women [91,92]. Because we did not control for the presence of anxiety and depression in our study, we can not distinguish between the effects of emotional dysregulation strategies (including different forms of negative and

 

positively dysregulated affect like irritability, anger, and sensation seeking) and the effects of mood disorders in the relationship with ADHD symptoms and executive function deficits [93].

Fourth, because participants in our study were young adults (between 20–30 years of age), the associations between ADHD and emotional dysregulation may not hold in other age groups. Given evidence that emotional dysregulation tends to decrease with age in individuals with ADHD [34], there is a need to examine our research questions in larger samples and in other age groups. Finally, the cross-sectional design of the study does not allow us to draw a causal relationship between the variables. While the proposed theoretical model, according to which executive function deficits mediate the relationship between ADHD and emotional dysregulation, was supported by the literature [94], alternative models exist. For example, Marques et al. [77] found that emotion dysregulation fully mediated the relationship between executive function deficits (i.e., inhibitory control difficulties) and aggressive behavior in ADHD children.

## Conclusions

While previous research investigating the relationship between ADHD and emotional dysregulation is limited by a focus on male participants [95], the current study provides insight into the centrality of emotional regulation deficits in women with ADHD and their associations with different types of executive function deficits. It shows that emotional dysregulation is predicted by ADHD symptoms severity and that executive function deficits might be a possible underlying mechanism of this link.

Although emotional dysregulation is considered a transdiagnostic condition, contributing to many types of psychopathologies in adults and adolescents [96,97], shifting clinical attention to emotional dysregulation symptoms may offer valuable theoretical and clinical implications for the diagnosis and treatment of women's ADHD. Several studies suggested that ADHD with emotional dysregulation may represent a distinct profile of ADHD characterized by more severe and pervasive symptoms, higher levels of functional impairment, higher comorbidity, and more severe executive function deficits, and is only partially responsive to treatments for ADHD [81,98]. Therefore, addressing emotional dysregulation may assist in accurate diagnosis and treatment planning. Faraone et al. [81] also emphasized the importance of developing a new assessment tool that effectively captures the multidimensional aspects of emotional symptoms in ADHD. Such a measure would aid in identifying emotional symptoms within the context of ADHD, distinguishing ADHD from other disorders, and enhancing the evaluation of treatment-related changes. Emotional dysregulation in ADHD may also benefit from targeted pharmacological and psychosocial interventions [Baweja et al., 2022]. Previous studies have shown that psychosocial interventions, including cognitive behavioral therapy, mindfulness, or family interventions for emotional symptoms, improved the overall functioning of children with ADHD [99], adolescents [100], and adults [101]. Moreover, while studies support the use of ADHD medications for reducing emotional dysregulation in ADHD patients [102], research has also suggested that ADHD medications may be less effective on bottom-up mechanisms underlying emotional dysregulation [103]. Therefore, the use of antidepressants and/or mood stabilizers should be considered for improving emotional regulation instead of ADHD medications [103]. Finally, monitoring developmental trajectories of emotion regulation could help flag at-risk girls and provide a promising transdiagnostic preventive intervention [104].

### Public health significance

• Women's ADHD is related to executive function deficits and emotional dysregulation.

• Emotional dysregulation is a core component of ADHD.

• Assessment and diagnosis of women's ADHD should address problems of emotional dysregulation.

## Supporting information

**S1 Text. Tables and figures.**
(DOCX)

**S1 File. Women adhd_file.**
(XLS)

## Author contributions

**Conceptualization:** Ortal Slobodin, Michal Har Sinay, Ada H. Zohar.

**Data curation:** Michal Har Sinay.

**Formal analysis:** Michal Har Sinay.

**Investigation:** Ortal Slobodin.

**Methodology:** Ortal Slobodin, Michal Har Sinay, Ada H. Zohar.

**Project administration:** Michal Har Sinay.

**Supervision:** Ada H. Zohar.

**Validation:** Ortal Slobodin.

**Writing – original draft:** Ortal Slobodin.

**Writing – review & editing:** Michal Har Sinay, Ada H. Zohar.

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
