## [Decision Letter · Decision Letter 0]

5 Aug 2025

Dear Dr. Slobodin,

Thank you for submitting your manuscript to PLOS ONE. After careful consideration, we feel that it has merit but does not fully meet PLOS ONE’s publication criteria as it currently stands. Therefore, we invite you to submit a revised version of the manuscript that addresses the points raised during the review process.

We look forward to receiving your revised manuscript.

Kind regards,

Lu Hua Chen, PhD, M.D.

Academic Editor

PLOS ONE

Journal Requirements:

2. Please describe in your methods section how capacity to provide consent was determined for the participants in this study. Please also state whether your ethics committee or IRB approved this consent procedure. If you did not assess capacity to consent please briefly outline why this was not necessary in this case.

- https://doi.org/10.1007/s11920-024-01492-6

In your revision ensure you cite all your sources (including your own works), and quote or rephrase any duplicated text outside the methods section. Further consideration is dependent on these concerns being addressed.

6. We are unable to open your Supporting Information file [data file.sav]. Please kindly revise as necessary and re-upload.

Reviewers' comments:

Reviewer's Responses to Questions

**Comments to the Author**

1. Is the manuscript technically sound, and do the data support the conclusions?

Reviewer #1: Yes

Reviewer #2: Yes

Reviewer #3: Yes

Reviewer #4: Partly

2. Has the statistical analysis been performed appropriately and rigorously?

Reviewer #1: Yes

Reviewer #2: I Don't Know

Reviewer #3: Yes

Reviewer #4: Yes

3. Have the authors made all data underlying the findings in their manuscript fully available?

Reviewer #1: Yes

Reviewer #2: Yes

Reviewer #3: Yes

Reviewer #4: Yes

4. Is the manuscript presented in an intelligible fashion and written in standard English?

Reviewer #1: Yes

Reviewer #2: Yes

Reviewer #3: Yes

Reviewer #4: Yes

Reviewer #1: The manuscript addresses a pertinent issue related to emotional dysregulation in adult women with ADHD. The manuscript is structured. The study is grounded in relevant theoretical models and supported by a thorough review of literature. The use of validated scales such as ASRS, DERS, BRIEF-A alongwith appropriate statistical analyses viz. hierarchical regressions etc. strengthens the methodological rigor of this study.

Reviewer #2: I think the paper is relevant, about a relevant topic and still not sufficiently researched. It is well written and organized. However, the following features should be improved.

Introduction:

P.5, line 115: “The second model perceives ADHD with emotional dysregulation as qualitatively different from pure ADHD, which has a distinct etiological entity and ADHD course”.- There is no reference for this model.

Methods:

P.9, line 205: “All participants signed a written informed consent”. – By the position this statement is in the text, it seems only the first group (women with ADHD) signed this consent.

P.10, line 225: “The three groups were found to have very similar characteristics, and were not significantly different in age, years of education and employment”.- The variable “employment” is not present in the preious sample characterization, the mean age and SD are also absent…a table with all the information about the sample should be presented.

P.12, line 285: “In the current study, the reliability of the subscales was .88, .80, and .59…”.- this last value is not acceptable, and the authors have already removed a subscale of the Emotion Regulation scale due to a value of .57. I think this should be removed too.

P.13, line 296: Data Analysis: The analysis on the normality of distribution and homogeneity of variance should be presented.

P.13, line 298: “To examine the second hypothesis, we conducted a Univariate Analysis of Variance with post-hoc comparisons using the Bonferroni range correction”. – why didn´t the authors use MANOVA? That’s the adequate analysis for multiple dependent variables and protects against Type I errors; ANOVA has Bonferroni correction for post-hoc tests only, but not for the initial composite.

Discussion:

The authors do not discuss results that were not expected. Namely, the fact that positive affect does not correlate with ADHD symptoms severity, and that there are no significant differences in positive affect between Groups 1 and 3.

P.16, line 370: “emotional liability” – typo.

P.17, limitations: there is, in my view, a major limitation that the authors did not state: the executive functions should not be accessed by self-report, but by performance-based tests.

P.17, line e 410: “Third, reliance on self-report measures may involve problems of shared method variance”.- The authors can assess this with statistical methods, I recommend this.

P.18, line 427: Shifting clinical attention to emotional dysregulation symptoms, rather than core ADHD symptoms of hyperactivity and impulsivity which are usually less dominant in women, may improve identification and intervention”.- I agree with the fact that it may improve intervention. However, how can it improve identification of ADHD? By itself, it can´t, because emotional dysregulation is present in several disorders…; the authors should reflect on this and reframe.

Reviewer #3: This manuscript presents an important contribution to the field of ADHD research, particularly focusing on its manifestation in females. The authors have conducted a comprehensive study exploring the prevalence, symptomatology, and executive functioning aspects of ADHD in female populations. While the overall structure and content of the manuscript are well presented, the paper would benefit from a review for minor topographical errors (e.g. line 303 should read a-priori) and there are some areas that could benefit from revision to enhance its clarity and impact.

The authors provide a thorough background on ADHD, but there are opportunities to strengthen the presentation of current data. The prevalence statistics cited are predominantly from 2012, which is now over a decade old. It would significantly enhance the paper's relevance to update these figures with more recent epidemiological data on ADHD prevalence, especially in females. This update would provide a more accurate representation of the current landscape of ADHD diagnosis and prevalence.

A key strength of this manuscript is its focus on ADHD in females, an often-underrepresented group in ADHD research, and differences in terms of gender-specific manifestations are included. However, the discussion on potential differences in ADHD types in females could be expanded slightly, by elaborating on whether females tend to exhibit more inattentive or internalizing symptoms. This exploration could provide valuable insights into why ADHD might be less frequently detected in females, possibly due to inattentive symptoms being less outwardly visible.

The theoretical framework of the study is well-constructed, but there are areas that require clarification. For instance, the second theoretical model mentioned on line 115 lacks a clear reference. Adding this reference would strengthen the theoretical underpinnings of the study. Additionally, while the authors provide a comprehensive discussion on emotional regulation and executive functioning, it would be beneficial to offer a more explicit rationale earlier in the section for choosing shifting and working memory as the two executive functioning components to examine. This would help readers better understand the study's focus and its potential implications.

The methodology of the study is generally sound, although there are some concerns regarding sampling selection. It is commendable that the authors acknowledge these limitations in the discussion section, demonstrating a critical awareness of the study's constraints. Given another limitation of the study is the use of self- report measures, it would be beneficial to include a discussion on the limitations of self-report measures and any steps taken to mitigate their influence

Reviewer #4: This manuscript presents a well-designed and timely investigation into emotional dysregulation in adult women with ADHD. The paper is generally well-structured and clearly written, with appropriate statistical methods and transparent reporting. However, there are several issues that require clarification or revision before the manuscript can be considered for publication. The study relies on self-reported ADHD diagnosis and ASRS cut-off scores. While this is common in large-scale surveys, it raises concerns about diagnostic validity. Please clarify whether any clinical interviews or verification methods were used to confirm diagnoses. If not, consider discussing in more depth the limitations of this approach.

The creation of a “probable ADHD” group is innovative but may introduce interpretative complexity. Please elaborate on the rationale for including this group in primary analyses and whether statistical validation supports treating it as a separate category.

Emotional dysregulation often co-occurs with depression and anxiety, both of which were not controlled for in your analysis. This is a significant limitation, particularly given the centrality of emotional symptoms in your outcome measures. Please address this issue in the discussion and consider its implications for interpreting your results.

The CERI scale is a promising ADHD-specific tool, but you note low reliability of subscales. Was the total score validated independently? Please clarify why it was used in spite of the reliability issues and consider discussing the implications of this choice. The mediation analyses are well-executed using Hayes’ PROCESS macro. However, it would strengthen the manuscript to report indirect effect sizes (e.g., standardized β) and explicitly state whether the mediation is full or partial.

Minor issue: Line 29: Likely a typo—please confirm if “emotional liability” should be “emotional lability.”

**Do you want your identity to be public for this peer review?** For information about this choice, including consent withdrawal, please see our Privacy Policy

Reviewer #1: **Yes: ** Dr. Syed Sajid Husain Kazmi

Reviewer #2: No

Reviewer #3: No

Reviewer #4: No

---

## [Author Response · Author response to Decision Letter 1]

11 Sep 2025

We would like to express our deep gratitude to the editor and the reviewers for the comprehensive reading and constructive comments. We found them highly relevant and useful in improving the manuscript. Below is a point-by-point response to the reviewers' comments. All comments were fully addressed. The changes made in the manuscript are highlighted in yellow.

Responses to reviewers' comments:

Reviewer #2:

Comment 1: Introduction: P.5, line 115: “The second model perceives ADHD with emotional dysregulation as qualitatively different from pure ADHD, which has a distinct etiological entity and ADHD course”.- There is no reference for this model.

Response to comment 1: A reference was added (page 6, lines 127-129).

Comment 2: Methods: P.9, line 205: “All participants signed a written informed consent”. – By the position this statement is in the text, it seems only the first group (women with ADHD) signed this consent.

Response to comment 2: Thank you for this note. We have moved this statement to the end of the paragraph so it will reflect the fact that all participants signed an informed consent (page 10, line 223).

Comment 3: P.10, line 225: “The three groups were found to have very similar characteristics, and were not significantly different in age, years of education and employment”.- The variable “employment” is not present in the previous sample characterization, the mean age and SD are also absent…a table with all the information about the sample should be presented.

Response to comment 3: As suggested, a table describing the background variables of the three study groups was added (Table 1). The text was adjusted accordingly (page 11, lines 252-258). We also added information about the background variables we collected to the "measures" section (page 11, line 261-262).

Comment 4: P.12, line 285: “In the current study, the reliability of the subscales was .88, .80, and .59…”.- this last value is not acceptable, and the authors have already removed a subscale of the Emotion Regulation scale due to a value of .57. I think this should be removed too.

Response to comment 4: Thank you for this note. Following this suggestion, we removed the nine items included in the externally-oriented thinking subscale from our analyses. We addressed this decision in the text. We also reported on the total reliability of the Alexithymia scale after removing the externally-oriented thinking subscale ( .90). (page 13, lines 310-314).

Comment 5: P.13, line 296: Data Analysis: The analysis on the normality of distribution and homogeneity of variance should be presented.

Response to comment 5: Information about normality and homogeneity was added to the text (Table 2, and in the text- page 14, lines 336-340).

Comment 6: P.13, line 298: “To examine the second hypothesis, we conducted a Univariate Analysis of Variance with post-hoc comparisons using the Bonferroni range correction”. – why didn´t the authors use MANOVA? That’s the adequate analysis for multiple dependent variables and protects against Type I errors; ANOVA has Bonferroni correction for post-hoc tests only, but not for the initial composite.

Response to comment 6: Thank you for this comment. We have replaced the ANOVA with MANOVA (pages 15-16, lines 355-371).

Comment 7: Discussion: The authors do not discuss results that were not expected. Namely, the fact that positive affect does not correlate with ADHD symptoms severity, and that there are no significant differences in positive affect between Groups 1 and 3.

Response to comment 7: Following this comment, we added a new paragraph to the discussion where we discuss these unexpected findings and provide several possible explanations, including the structure of the PANAS, positive illusory bias, and differences between ADHD and non-ADHD women in experiencing and expressing negative emotions (pages 19-20, lines 445-470).

Comment 8: P.16, line 370: “emotional liability” – typo.

Response to comment 8: This error was corrected (page 18, line 421).

Comment 9: P.17, limitations: there is, in my view, a major limitation that the authors did not state: the executive functions should not be accessed by self-report, but by performance-based tests.

Response to comment 9: We agree that the exclusive reliance on self-report measures, especially in assessing executive function deficits, is a major limitation of our study. To address this shortcoming, we added a new part to the limitations section that elaborates on this point (page 20, line 471-488).

Comment 10: P.17, line e 410: “Third, reliance on self-report measures may involve problems of shared method variance”.- The authors can assess this with statistical methods, I recommend this.

Response to comment 10: As suggested, we tested for shared method variance. Variance Inflation Factor (VIF) values indicated no issues with multicollinearity, with all predictors showing VIFs below 2.5. Specifically, VIFs ranged from 1.03 to 2.47. This point was clarified in the text (pages 14-15, lines 340-342).

Comment 11: P.18, line 427: Shifting clinical attention to emotional dysregulation symptoms, rather than core ADHD symptoms of hyperactivity and impulsivity which are usually less dominant in women, may improve identification and intervention”.- I agree with the fact that it may improve intervention. However, how can it improve identification of ADHD? By itself, it can´t, because emotional dysregulation is present in several disorders…; the authors should reflect on this and reframe.

Response to comment 11: Following this note, we have rephrased this argument, suggesting that "Although emotional dysregulation is considered a transdiagnostic condition, contributing to many types of psychopathologies in adults and adolescents (Fernandez et al., 2016; Kring, 2008), shifting clinical attention to emotional dysregulation symptoms may offer valuable theoretical and clinical implications for the diagnosis and treatment of women's ADHD." In this part, we also elaborate on the different benefits of addressing emotional dysregulation in ADHD, including in the context of psychosocial and pharmacological interventions (pages 22-23, lines 516-541).

Reviewer #3:

Comment 1: This manuscript presents an important contribution to the field of ADHD research, particularly focusing on its manifestation in females. The authors have conducted a comprehensive study exploring the prevalence, symptomatology, and executive functioning aspects of ADHD in female populations. While the overall structure and content of the manuscript are well presented, the paper would benefit from a review for minor typographical errors (e.g. line 303 should read a-priori) and there are some areas that could benefit from revision to enhance its clarity and impact.

Response to comment 1: Thank you for the positive feedback. The manuscript has now undergone a thorough review to remove typos and ensure clarity.

Comment 2: The authors provide a thorough background on ADHD, but there are opportunities to strengthen the presentation of current data. The prevalence statistics cited are predominantly from 2012, which is now over a decade old. It would significantly enhance the paper's relevance to update these figures with more recent epidemiological data on ADHD prevalence, especially in females. This update would provide a more accurate representation of the current landscape of ADHD diagnosis and prevalence.

Response to comment 2: As suggested, we have updated the epidemiological data on ADHD prevalence and male-female ratio, including data from 2023-2025 (page 3, lines 62-66).

Comment 3: A key strength of this manuscript is its focus on ADHD in females, an often-underrepresented group in ADHD research, and differences in terms of gender-specific manifestations are included. However, the discussion on potential differences in ADHD types in females could be expanded slightly, by elaborating on whether females tend to exhibit more inattentive or internalizing symptoms. This exploration could provide valuable insights into why ADHD might be less frequently detected in females, possibly due to inattentive symptoms being less outwardly visible.

Response to comment 3: As suggested, we have now elaborated the discussion about ADHD types in women in the context of under-identification and diagnosis (page 4, lines 84-89).

Comment 4: The theoretical framework of the study is well-constructed, but there are areas that require clarification. For instance, the second theoretical model mentioned on line 115 lacks a clear reference. Adding this reference would strengthen the theoretical underpinnings of the study.

Response to comment 4: A reference was added (page 6, lines 127-129).

Comment 5: Additionally, while the authors provide a comprehensive discussion on emotional regulation and executive functioning, it would be beneficial to offer a more explicit rationale earlier in the section for choosing shifting and working memory as the two executive functioning components to examine. This would help readers better understand the study's focus and its potential implications.

Response to comment 5: To provide a clearer rationale for focusing on working memory and task shifting as potential mediators of the ADHD-emotional dysregulation link, we have rewritten the introduction part in the beginning of the manuscript (page 3, lines 43-54).

Comment 6: The methodology of the study is generally sound, although there are some concerns regarding sampling selection. It is commendable that the authors acknowledge these limitations in the discussion section, demonstrating a critical awareness of the study's constraints. Given another limitation of the study is the use of self- report measures, it would be beneficial to include a discussion on the limitations of self-report measures and any steps taken to mitigate their influence.

Response to comment 6: As suggested, we addressed the sampling limitations, which include a relatively limited sample of non-referred, educated women who volunteered to participate in the study and may not represent women with more extreme levels of ADHD symptoms and functional impairment (Page 20, lines 470-475). We also addressed the problems involved in self-report measures (page 20, lines 480-484) and described the strategies we used to reduce them.

Reviewer #4:

Comment 1: This manuscript presents a well-designed and timely investigation into emotional dysregulation in adult women with ADHD. The paper is generally well-structured and clearly written, with appropriate statistical methods and transparent reporting. However, there are several issues that require clarification or revision before the manuscript can be considered for publication. The study relies on self-reported ADHD diagnosis and ASRS cut-off scores. While this is common in large-scale surveys, it raises concerns about diagnostic validity. Please clarify whether any clinical interviews or verification methods were used to confirm diagnoses. If not, consider discussing in more depth the limitations of this approach.

Response to comment 1: Following this comment (as well as the comments of the previous reviewers), we described the limited validity of self-report measures of ADHD (both self-report diagnosis and ASRS) as early as the methods section (pages 10-11, lines 231-244). In the limitation section, we further addressed the problems involved in self-report measures (page 20, lines 470-484).

Comment 2: The creation of a “probable ADHD” group is innovative but may introduce interpretative complexity. Please elaborate on the rationale for including this group in primary analyses and whether statistical validation supports treating it as a separate category.

Response to comment 2: The clinical and empirical rationale for creating the three groups, as well as statistical analysis that supports this distinction, were added to the methods section (pages 10-11, lines 231-244).

Comment 3: Emotional dysregulation often co-occurs with depression and anxiety, both of which were not controlled for in your analysis. This is a significant limitation, particularly given the centrality of emotional symptoms in your outcome measures. Please address this issue in the discussion and consider its implications for interpreting your results.

Response to comment 3: This limitation is now explicitly addressed (pages 20-21, lines 487-495).

Comment 4: The CERI scale is a promising ADHD-specific tool, but you note low reliability of subscales. Was the total score validated independently? Please clarify why it was used in spite of the reliability issues and consider discussing the implications of this choice.

Response to comment 4: Given the low reliability of the scale and undesirable Skewness and Kurtosis values indicating a leptokurtic distribution, we decided to remove this scale from our analysis.

Comment 5: The mediation analyses are well-executed using Hayes’ PROCESS macro. However, it would strengthen the manuscript to report indirect effect sizes (e.g., standardized β) and explicitly state whether the mediation is full or partial.

Response to comment 5: Standardized β values were added to the text. We also reported about the partial mediation results (page 17, lines 390-409).

Comment 6: Minor issue: Line 29: Likely a typo—please confirm if “emotional liability” should be “emotional lability.”

Response to comment 6: This error was corrected (page 18, line 420).

---

## [Decision Letter · Decision Letter 1]

9 Nov 2025

A Controlled Study of Emotional Dysfunction in Adult Women with ADHD

PONE-D-25-21388R1

Dear Dr. Slobodin,

We’re pleased to inform you that your manuscript has been judged scientifically suitable for publication and will be formally accepted for publication once it meets all outstanding technical requirements.

Kind regards,

Lu Hua Chen, PhD, M.D.

Academic Editor

PLOS ONE

Additional Editor Comments (optional):

Reviewers' comments:

Reviewer's Responses to Questions

**Comments to the Author**

Reviewer #2: All comments have been addressed

Reviewer #3: All comments have been addressed

Reviewer #4: All comments have been addressed

2. Is the manuscript technically sound, and do the data support the conclusions?

Reviewer #2: (No Response)

Reviewer #3: Yes

Reviewer #4: Yes

3. Has the statistical analysis been performed appropriately and rigorously?

Reviewer #2: (No Response)

Reviewer #3: Yes

Reviewer #4: Yes

4. Have the authors made all data underlying the findings in their manuscript fully available?

Reviewer #2: (No Response)

Reviewer #3: Yes

Reviewer #4: Yes

5. Is the manuscript presented in an intelligible fashion and written in standard English?

Reviewer #2: (No Response)

Reviewer #3: Yes

Reviewer #4: Yes

Reviewer #2: (No Response)

Reviewer #3: (No Response)

Reviewer #4: I have reviewed the authors’ responses and revisions. I accept the comments provided and have no further questions or additional concerns regarding the manuscript.

**Do you want your identity to be public for this peer review?** For information about this choice, including consent withdrawal, please see our Privacy Policy

Reviewer #2: No

Reviewer #3: No

Reviewer #4: No

---

## [Editor Report · Acceptance letter]

PONE-D-25-21388R1

PLOS ONE

Dear Dr. Slobodin,

I'm pleased to inform you that your manuscript has been deemed suitable for publication in PLOS ONE. Congratulations! Your manuscript is now being handed over to our production team.

Kind regards,

on behalf of

Dr. Lu Hua Chen

Academic Editor

PLOS ONE